# T cell-specific inhibition of multiple apoptotic pathways blocks negative selection and causes autoimmunity

Megan L Burger, Kenneth K Leung, Margaux J Bennett, Astar Winoto*

Department of Molecular and Cell Biology, University of California, Berkeley, Berkeley, United States

**Abstract** T cell self-tolerance is thought to involve peripheral tolerance and negative selection, involving apoptosis of autoreactive thymocytes. However, evidence supporting an essential role for negative selection is limited. Loss of Bim, a Bcl-2 BH3-only protein essential for thymocyte apoptosis, rarely results in autoimmunity on the C57BL/6 background. Mice with T cell-specific over-expression of Bcl-2, that blocks multiple BH3-only proteins, are also largely normal. The nuclear receptor Nur77, also implicated in negative selection, might function redundantly to promote apoptosis by associating with Bcl-2 and exposing its potentially pro-apoptotic BH3 domain. Here, we report that T cell-specific expression of a *Bcl2* BH3 mutant transgene results in enhanced rescue of thymocytes from negative selection. Concomitantly, $T_{reg}$ development is increased. However, aged BH3 mutant mice progressively accumulate activated, autoreactive T cells, culminating in development of multi-organ autoimmunity and lethality. These data provide strong evidence that negative selection is crucial for establishing T cell tolerance.

## Introduction

Maintenance of T cell self-tolerance is essential for the prevention of autoimmune diseases. A series of mechanisms are thought to be involved in establishing T cell tolerance, the first of which is deletion of potentially self-reactive T cells in the thymus. During T cell development, autoreactive T cells bind self-peptide presented on major histocompatibility complex (MHC) molecules with high affinity and receive strong T cell receptor (TCR) signals that trigger apoptosis and clonal deletion. Some of these cells can also differentiate down an alternative pathway to a harmless lineage, such as T regulatory cells ($T_{reg}$), or also CD8αα and invariant NK T cells (*Stritesky et al., 2012*). The importance of this process, termed negative selection, in establishing self-tolerance and preventing autoimmunity was best demonstrated in studies of mice deficient in the autoimmune regulator (*Aire*) gene. These mice develop a multi-organ autoimmune disease as the result of defective deletion of auto-reactive T cells specific to tissue-restricted antigens (TRA), which would normally be expressed by medullary thymic epithelial cells under the control of AIRE (*Anderson et al., 2002*). Humans with loss of AIRE function develop similar disease pathology (*Finnish-German APECED Consortium., 1997*; *Nagamine et al., 1997*). However, additional models that implicate a breakdown in negative selection as the driving force behind the development of autoimmune disease are lacking (*von Boehmer and Melchers, 2010*). This might suggest that under most circumstances, subsequent tolerizing mechanisms in the peripheral compartments such as anergy or $T_{reg}$-mediated suppression are sufficient to compensate for defective thymic deletion. Even in the case of AIRE-deficiency, impaired tolerization of CD4+ T cells in the periphery by extra-thymic AIRE expressing cells likely contributes to the development of autoimmunity (*Gardner et al., 2013*). This has cast doubt on the relative importance of negative selection for maintaining immunological self-tolerance.

*For correspondence: winoto@berkeley.edu

**Competing interests:** The authors declare that no competing interests exist.

**Reviewing editor**: Diane Mathis, Harvard Medical School, United States

**eLife digest** Our immune system protects us from disease by recognizing and mounting a defence against harmful pathogens that enter our bodies. T cells, a type of white blood cell, play a key role in this process. Each T cell has a unique protein called a T cell receptor on its surface that is able to recognize particular pieces of pathogens. Together, the millions of T cells in our bodies, each with its own unique T cell receptor, can initiate an immune response to eliminate a vast array of potential pathogens.

T cells are made in an organ called the thymus. During this production process, immature T cells are generated, including some with T cell receptors that recognize the harmless molecules that make up our bodies. If allowed to enter the bloodstream and left to their own devices, these T cells could trigger an immune response against the body, leading to the development of autoimmune disease.

Normally, many of these 'auto-reactive' T cells are instructed to die in the thymus by a process called negative selection. Furthermore, auto-reactive cells that escape into the blood can also be shut down by additional failsafe mechanisms. Given the success of these failsafe mechanisms, notably the effectiveness of a class of T cells called T regulatory cells, some researchers have begun to ask if negative selection is necessary to prevent autoimmunity.

During negative selection T cells die as a result of a process called apoptosis. Multiple proteins have been implicated in T cell apoptosis, including Bim, Puma and the Nur77 family of nuclear receptors. Blocking the function of some these proteins individually can rescue some autoreactive T cells from death: however, this rarely results in the development of autoimmune disease.

Burger et al. have now created a mouse strain with T cells that produce large amounts of a mutant form of the anti-apoptotic protein, Bcl-2, which can block the function of multiple pro-apoptotic proteins, including Bim and Puma. Additionally, it has been proposed that Bcl-2 can be converted to a pro-apoptotic protein by Nur77 proteins, but the Bcl-2 proteins in the mutant strain are able to resist this process.

T cells in the thymus of the mutant mice were highly resistant to apoptosis accompanying negative selection. Moreover, as the mice aged, they accumulated autoreactive T cells in the blood, which led to symptoms of autoimmune disease and early death. While various failsafe mechanisms were engaged, they did not provide sufficient protection. The work of Burger et al. thus provides strong evidence that negative selection in the thymus is crucial for the prevention of autoimmune disease.

Apoptosis of thymocytes during negative selection is dependent on the mitochondrial apoptosis pathway regulated by the Bcl-2 family of proteins (*Tischner et al., 2010*). In particular, the pro-apoptotic, Bcl-2 homology domain 3 (BH3) only protein, Bim, which is upregulated in response to strong TCR signals, is essential for this process. Bim and other BH3-only proteins are thought to promote apoptosis by binding and inhibiting anti-apoptotic Bcl-2 proteins (such as Bcl-2, Bcl-X$_L$ and Mcl-1) that directly suppress downstream effectors Bax and Bak, which are responsible for disrupting mitochondrial membrane integrity (*Youle and Strasser, 2008*). Bim-deficiency has been shown to cause defective thymocyte apoptosis in several TCR transgenic (HY, OTI, OTII) models of negative selection (*Bouillet et al., 2002*; *Moran et al., 2011*; *Zhan et al., 2011*; *Suen and Baldwin, 2012*). However, this apoptotic defect does not lead to emergence of autoreactive mature T cells except in the case of negative selection to TRA (*Hu et al., 2009*; *Kovalovsky et al., 2010*; *Suen and Baldwin, 2012*). Interestingly, *Bim$^{-/-}$* mice on a mixed 129/Sv X C57BL/6 background develop a systemic lupus erythematosus (SLE)-like autoimmune disease, indicative of a breakdown in tolerance (*Bouillet et al., 1999*). However, loss of Bim in other immune compartments likely contributes to the disease, especially B cells, which are important mediators of SLE pathology. This phenotype was greatly ameliorated on the autoimmune-resistant C57BL/6 background and is distinct from the multi-organ, T cell dominant disease found in AIRE-deficient mice (*Bouillet et al., 2001*; *Labi et al., 2014*). Recently, a study showed that additional loss of Puma (*Bbc3*), another BH3-only protein, enhanced the *Bim$^{-/-}$* thymic deletion defect and led to the development of immune pathology more similar to that found in AIRE-deficient mice (*Gray et al., 2012*). T cells from these mice were able to transfer the disease, lending support to

it being T cell-driven. However, the role of non-T cells in these mice cannot be excluded due to germline deletion of Puma and Bim. In addition, T-cell specific over-expression of Bcl-2, which can inhibit both Bim and Puma (*Chen et al., 2005*), does not lead to autoimmunity (*Sentman et al., 1991*; *Linette et al., 1995*), suggesting that a defect in T cells alone may be insufficient to cause disease. Alternatively, redundant negative selection pathways involving Bim, Puma and the Nur77 family members (See below) may not allow Bcl-2 over-expression to block all pathways leading to negative selection.

In addition to Bim, the Nur77 family of orphan steroid receptors, which includes Nur77, Nor-1 and Nurr1, has also been implicated in apoptosis accompanying negative selection. Nur77 expression, like that of Bim, is induced by strong TCR signals that result in negative selection. Low expression of both proteins has been correlated with defective clonal deletion in Non-obese Diabetic (NOD) mice (*Sohn et al., 2003*; *Liston et al., 2004*). Furthermore, T cell-specific over-expression of Nur77, or Nor-1, results in massive apoptosis of thymocytes (*Cheng et al., 1997b*). Expression of a dominant negative Nur77 protein that can block all family members results in inhibition of apoptosis in the F5 and HY TCR transgenic models of negative selection (*Calnan et al., 1995*; *Zhou et al., 1996*). Deficiency in Nur77 alone ($Nr4a1^{-/-}$) results in moderate inhibition of apoptosis accompanying negative selection in the OTII and BDC2.5 TCR transgenic models (*Fassett et al., 2012*). Interestingly, the Nur77 family has also been shown to transcriptionally regulate *Foxp3* expression and deletion of all three Nur77 family members ($Nr4a3^{-/-}$ X Cd4-$Cre^+Nr4a1^{fl/fl}Nr4a2^{fl/fl}$) results in impaired $T_{reg}$ cell production and development of a quick onset, severe autoimmune disease resembling that of Foxp3-deficient mice (*Sekiya et al., 2013*). Therefore, Nur77 could contribute to the maintenance of immunological tolerance both by facilitating deletion of autoreactive thymocytes and ensuring production of $T_{reg}$ cells. The mechanism by which Nur77 and its family members promote thymocyte apoptosis has long been under investigation. We and others showed that strong TCR signals induce the translocation of Nur77 and Nor-1 from the nucleus to the mitochondria in thymocytes, a phenomenon first reported to occur in cancer cells in response to apoptotic stimuli (*Lin et al., 2004*; *Thompson and Winoto, 2008*; *Wang et al., 2009*). At the mitochondria, Nur77 and Nor-1 bind Bcl-2 and cause exposure of Bcl-2's BH3 domain, the 'death' domain of the Bcl-2 family of proteins. Bcl-2 BH3 exposure correlates with thymocyte apoptosis in TCR-stimulated thymocytes and HY and F5 TCR transgenic thymocytes undergoing negative selection (*Thompson and Winoto, 2008*). BH3-exposed Bcl-2 might act similarly to BH3-only protein Bim to bind and inhibit other anti-apoptotic Bcl-2 proteins, such as Bcl-$X_L$, as was demonstrated in cancer cells (*Kolluri et al., 2008*). It is thus possible that Nur77 converts Bcl-2 to a pro-apoptotic effector during negative selection, preventing over-expression of Bcl-2 from completely blocking negative selection.

Here, we generated a T cell-specific *Bcl2* BH3 mutant transgenic mouse, in which Bcl-2's purported pro-apoptotic BH3 function should be abolished by changing its conserved BH3 residues, GDD, to alanines (*Cheng et al., 1997a*). We found that over-expression of both wild-type and BH3 mutant Bcl-2 efficiently rescued thymocyte apoptosis in two TCR transgenic models of negative selection. However, the BH3 mutant transgene more effectively blocked TCR-induced thymocyte apoptosis in vitro and better rescued high affinity TCR clones from deletion in polyclonal systems in vivo. Interestingly, BH3 mutant transgenic mice, in contrast to the reported wild-type *Bcl2* transgenic mouse phenotype, developed multi-organ autoimmune pathology and died around one year-of-age. Thus, we provide strong evidence that a breakdown in thymocyte apoptosis during negative selection is indeed sufficient to cause autoimmune disease.

## Results

### Generation of T cell-specific *Bcl2* BH3 domain mutant transgenic mice

To investigate the role of the Bcl-2 BH3 domain in thymocyte apoptosis, we created T cell-specific *Bcl2* BH3 mutant transgenic mice. Three amino acid residues critical for BH3 domain pro-apoptotic function were mutated to alanine (*Figure 1A*) and the BH3 mutated human *Bcl2* transgene (referred to as BH3) was expressed under the control of the *Cd4* regulatory elements (*Adlam and Siu, 2003*; *Xue et al., 2010*). The 'BH3' transgenic mice (BH3 Tg) were generated on the C57BL/6 background and two founder lines were chosen for analysis, BH3 A and BH3 B. For evaluation, the BH3 Tg mice were compared to the T cell-specific wild-type human Bcl-2 strain, $Lck^{Pr}$-*Bcl2*, also on the C57BL/6 background, referred to here as 'Bcl-2 Tg' (*Sentman et al., 1991*).

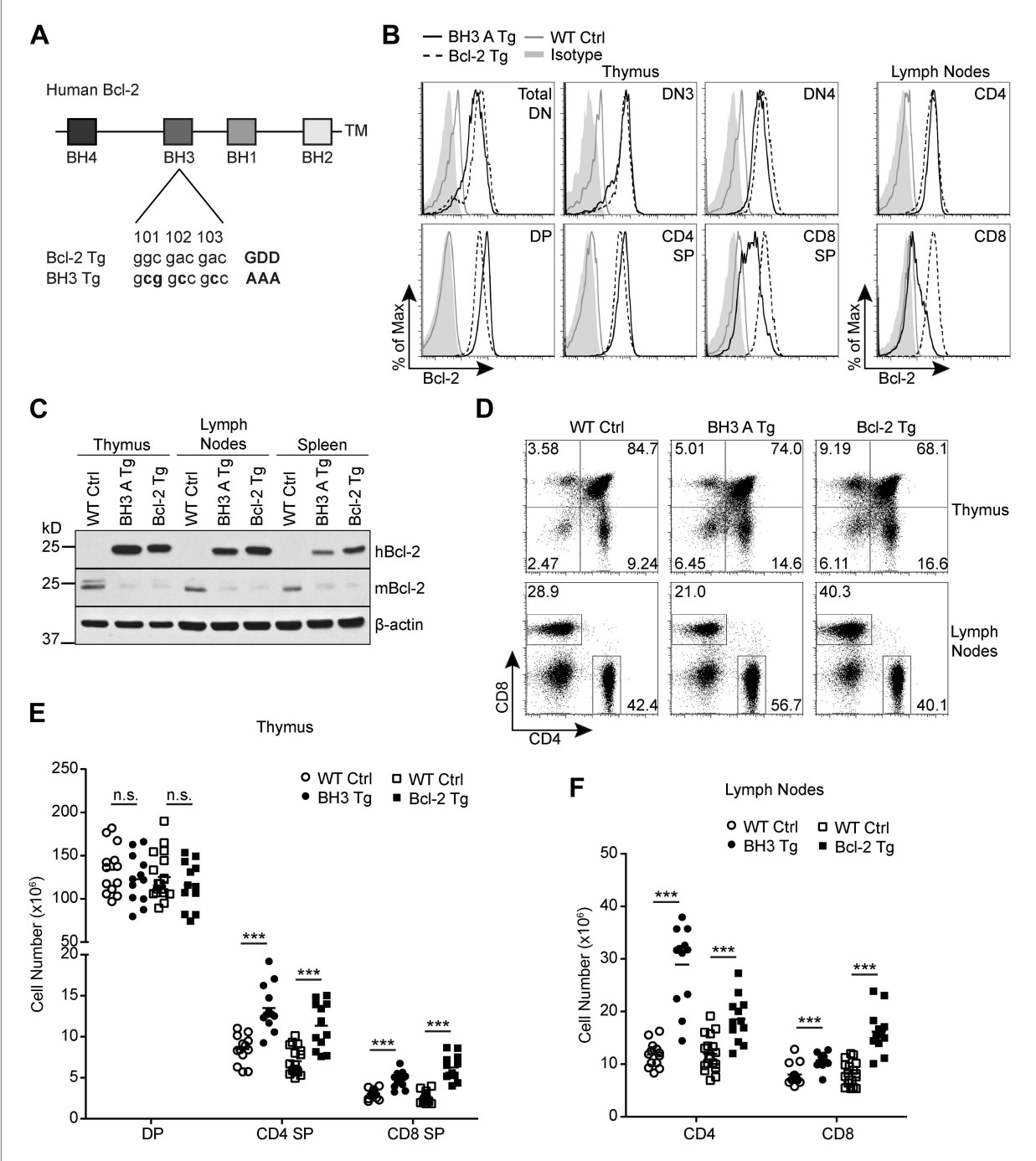

**Figure 1**. Generation of *Bcl2* BH3 domain mutant transgenic mice. (**A**) Diagram of the mutation made to the BH3 domain of human *Bcl2* to abolish BH3 domain function in the *Bcl2* BH3 mutant transgenic (BH3 Tg) mice. Amino acids 101–103 of human *Bcl2* were mutated from Glycine-Aspartate-Aspartate to Alanine-Alanine-Alanine to abolish BH3 domain function (***Cheng et al., 1997a***). (**B**) Flow cytometric analysis of intracellular Bcl-2 expression in BH3 Tg, line A (BH3 A Tg) vs *Lck^Pr-Bcl2* (Bcl-2 Tg) and wild-type (WT) thymocyte and mature T cell populations. DN populations were gated based on CD25 and CD44 expression. (**C**) Western blot analysis comparing human vs mouse Bcl-2 expression in the thymus, lymph nodes and spleen of BH3 A Tg vs Bcl-2 Tg and WT mice. (**D**) Flow cytometric analysis of CD4 vs CD8 T cell populations in the thymus and lymph nodes. (**E**) DP, CD4 SP and CD8 SP thymocyte cell numbers in 6-week-old BH3 Tg and Bcl-2 Tg mice compared to WT littermate controls. (**F**) Lymph node mature CD4 and CD8 T cell numbers in 6-week-old BH3 Tg and Bcl-2 Tg mice compared to WT littermate controls. The transgenic mice in (**B–D**) were age-matched within 1 week and compared to a littermate non-transgenic WT control. All mice were between 6 and 10-weeks-old. (**B–F**) are representative of or compiled from at least three independent experiments. Statistics here and in the following figures were calculated by Student's t-test: ***p < 0.001, **p < 0.01, *p < 0.05, n.s. not significant.

By intracellular staining with an antibody that detects both mouse and human Bcl-2, we observed that Bcl-2 expression was significantly increased in BH3 Tg thymocytes compared to wild-type controls (*Figure 1B*). Although the transgene is expressed by the *Cd4* promoter, increased Bcl-2 expression was detected in DN (CD4⁻ CD8⁻ double negative) thymocytes as early as CD25⁺ CD44⁻ DN3 (*Figure 1B*). This was very similar to the expression pattern of the *Lck*-driven wild-type *Bcl2* transgene. Additionally, the level of Bcl-2 upregulation in BH3 Tg vs Bcl-2 Tg thymocytes was very similar in DN, DP and CD4 single positive (CD4 SP) thymocytes. Only slightly higher expression of the BH3 transgene was noted in DP and CD4 SP cells. As expected, BH3 transgene expression was downregulated in CD8 single positive (CD8 SP) thymocytes due to loss of CD4 expression (*Figure 1B*). Lymph node CD8 T cells further downregulated BH3 transgene expression, while both BH3 Tg and Bcl-2 Tg CD4 T cells maintained similarly high levels (*Figure 1B*). Interestingly, we observed a compensatory downregulation of endogenous mouse Bcl-2 in the thymus, lymph nodes and spleen of both BH3 and Bcl-2 Tg mice (*Figure 1C*). Thus, we do not have to consider a contribution of endogenous mouse Bcl-2 to the phenotype of these mice. Due to close similarity in transgene expression level and pattern in the two BH3 Tg lines, BH3 A and BH3 B, we used these mice interchangeably throughout our study (Data not shown). In both BH3 Tg lines, we observed a decreased proportion of DP thymocytes and an increased proportion of SP thymocytes compared to wild-type littermate controls (*Figure 1D*). However, only the absolute SP cell numbers were increased, especially for CD4 SP cells, while the DP cell number was comparable to wild-type controls (*Figure 1E*). This is largely similar to the thymocyte distribution found in the wild-type Bcl-2 Tg mice (*Sentman et al., 1991*). As expected, the peripheral mature T cell compartment reflects the increased SP thymocyte numbers in both the BH3 Tg and Bcl-2 Tg mice (*Figure 1F*). Relative to wild-type mice, the CD4 to CD8 T cell ratio is therefore somewhat skewed in favor of CD4 T cells in the BH3 Tg mice and CD8 T cells in the Bcl-2 Tg mice. Hence, differential expression in the CD8 compartment is one caveat to comparison of BH3 and Bcl-2 transgenic mice. However, similar expression levels in the CD4 compartment should allow fair comparison of DP, CD4 SP and CD4 T cell phenotypes.

## Defective negative selection in BH3 Tg mice

A defect in thymocyte apoptosis accompanying negative selection could potentially contribute to increased SP thymocyte and mature T cell numbers in the BH3 and Bcl-2 Tg mice. However, while *Bcl2* transgene expression was previously shown to safeguard thymocytes against a variety of apoptotic stimuli, thymocytes were inefficiently protected from apoptosis during negative selection (*Sentman et al., 1991*; *Strasser et al., 1991*; *Bouillet et al., 2002*). To test whether BH3 Tg thymocytes might be better protected, we first assessed thymocyte apoptosis in response to strong TCR stimuli that mimic negative selection in vitro. In response to anti-CD3 and anti-CD28 antibody stimulation, BH3 transgene expression substantially decreased apoptosis of thymocytes compared to wild-type controls (*Figure 2A*). In addition, BH3 Tg thymocytes showed significantly increased resistance to apoptosis compared to Bcl-2 Tg thymocytes.

To test whether the BH3 transgene is also better able to block apoptosis during negative selection in vivo, we examined apoptosis of F5 TCR transgenic thymocytes in BH3 Tg mice crossed to F5 mice (BH3/F5) (*Mamalaki et al., 1992*, *1993*). We also examined Bcl-2/F5 mice for comparison. In response to injection with the F5 TCR-specific peptide, NP, F5 DP thymocytes undergo massive apoptosis (*Mamalaki et al., 1992*). Strikingly, however, expression of either the BH3 or *Bcl2* transgene almost completely inhibited thymocyte death, with thymocyte numbers only marginally decreased compared to PBS-injected control mice after 48 hr of treatment (*Figure 2B*). Analysis of thymocyte proportions indicated a significant rescue of the DP thymocyte population (*Figure 2C*). We sought to increase the sensitivity of the assay by using a previously described ex vivo thymic slice approach (*Dzhagalov et al., 2013*). This assay allows assessment of apoptosis over a more acute timeline, free of possible cytotoxic effects inherent in the NP injection model (*Mamalaki et al., 1992*). In this assay, transgenic thymic lobes were embedded in agarose, cut into 500 μm slices by vibratome and cultured in transwells. The slices were then briefly exposed to the relevant or an irrelevant control peptide and thymocyte apoptosis was assessed by flow cytometry at 12-hr intervals. Here, again, we saw near complete and equivalent protection of BH3 and Bcl-2 Tg thymocytes from apoptosis in response to NP peptide treatment over a 36-hr time course (*Figure 2D*). Thymocytes of all genotypes were similarly activated as measured by upregulation of CD69 (*Figure 2D*). We also performed an analogous thymic slice assay comparing apoptosis of thymocytes from female BH3 or Bcl-2 Tg mice crossed to HY TCR transgenic mice

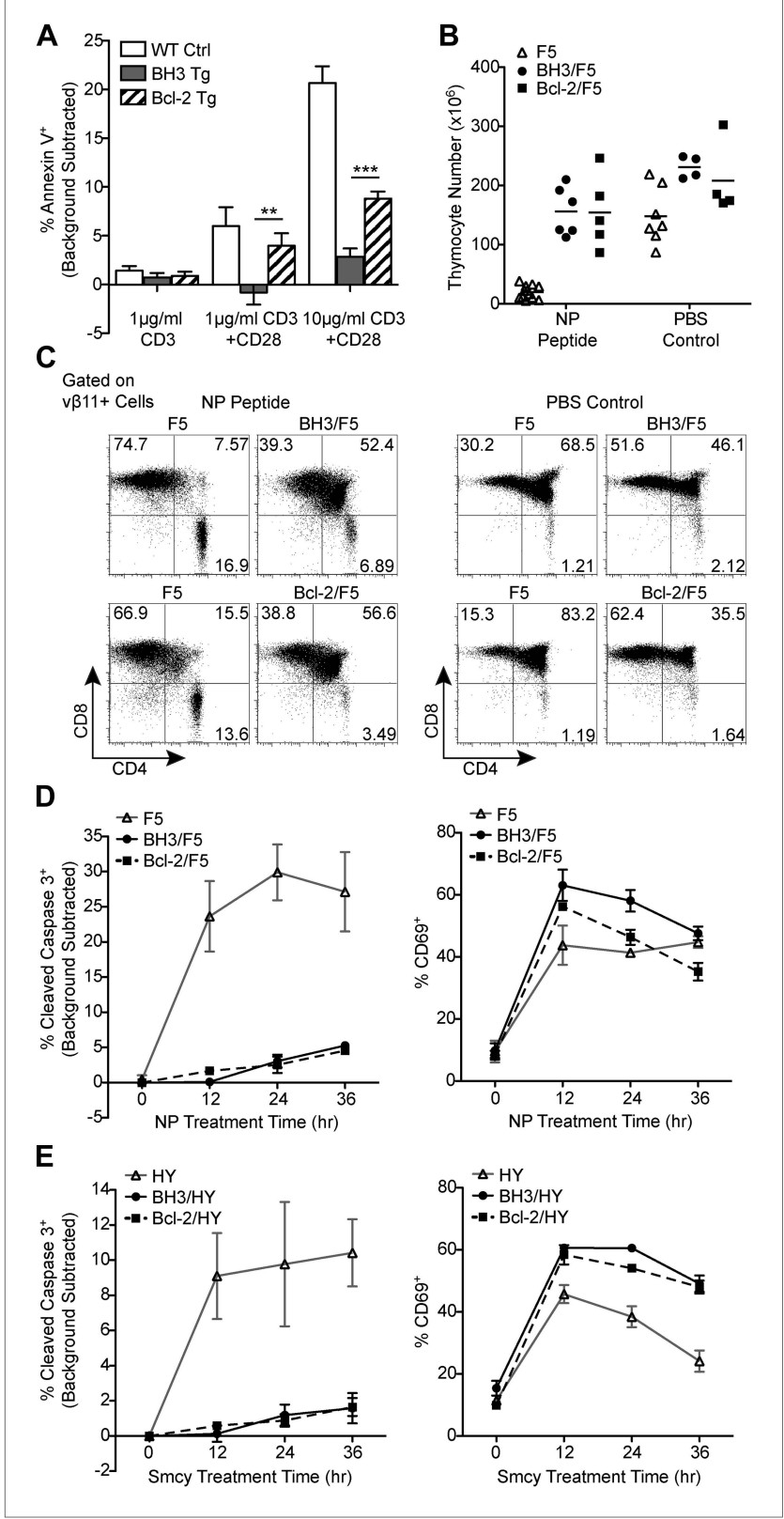

**Figure 2**. BH3 transgene expression efficiently blocks TCR-mediated thymocyte apoptosis in vitro and in F5 and HY TCR transgenic models of negative selection. (**A**) Thymocytes were stimulated for 18 hr with anti-CD3 and anti-CD28 (20 µg/ml) and analyzed for the percentage of Annexin V⁺ cells by flow cytometry. The percentage of
*Figure 2. Continued on next page*

*Figure 2. Continued*

Annexin V⁺ thymocytes in samples left untreated for 18 hr (i.e. background) was subtracted. (**B**) Total live thymocyte number in thymii collected 48 hr after injection with the F5 TCR-specific peptide, NP, or PBS. (**C**) CD4 vs CD8 flow cytometric analysis of F5 TCR⁺ (vβ11⁺) thymocytes from mice treated as in (**B**). Data in (**B** and **C**) are representative of or pooled from at least five independent experiments per transgenic genotype. Mice were between 6 and 9-weeks-old and F5 control mice were littermates to BH3/F5 and Bcl-2/F5 mice. (**D**) Percentage of Cleaved Caspase 3⁺ thymocytes by flow cytometric analysis treated with 100 ng/ml NP peptide for the indicated times. The percentage of Cleaved Caspase 3⁺ thymocytes treated with an irrelevant peptide (i.e. background) was subtracted at each time point. Efficient activation of thymocytes by the NP peptide was assessed by flow cytometric analysis of CD69 surface expression. Data are representative of three independent experiments. (**E**) Percentage of Cleaved Caspase 3⁺ thymocytes by flow cytometric analysis treated with 100 ng/ml of the HY TCR-specific peptide, Smcy, for the indicated times. Background was subtracted as in (**D**). Proper thymocyte activation was confirmed by analysis of CD69 expression. Data are representative of two independent experiments.

(BH3/HY or Bcl-2/HY, respectively) in response to the HY TCR-specific peptide, Smcy. Remarkably, just as in the F5 model, thymocytes expressing either transgene were significantly protected from apoptosis (*Figure 2E*). Hence, we conclude that Bcl-2 over-expression very efficiently inhibits thymocyte apoptosis accompanying negative selection in these models and this potent blockade could not be enhanced by expression of a BH3 mutant form of Bcl-2.

## BH3 transgene enhances rescue of high affinity TCR clones in a polyclonal system

We next sought to test whether over-expression of Bcl-2 is sufficient to block negative selection in a normal, polyclonal system and if in this context, the BH3 transgene might exhibit enhanced anti-apoptotic activity. To investigate this, we employed the Nur77GFP BAC transgenic reporter mouse system, in which GFP is inserted into the start site of the *Nr4a1* (Nur77) transgene (*Moran et al., 2011*). Studies of these mice have clearly demonstrated that the level of Nur77 expression in T cells directly correlates with the intensity of the TCR signals received (*Moran et al., 2011*). During thymocyte selection, cells receiving strong TCR signals are destined to die by negative selection while cells receiving very weak to no signal die by neglect. Hence, the post-selection thymocyte pool is largely comprised of cells expressing an intermediate level of GFP, reflecting their receipt of moderate, positively selecting TCR signals. In Bim⁻/⁻ X Nur77GFP mice, however, a significant number of post-selection thymocytes express high levels of GFP, indicating that high affinity TCR clones are rescued from negative selection in the absence of Bim (*Stritesky et al., 2013*). This rescue was also reflected by an increase in the total number of post-selection DP and SP thymocytes. When we assessed GFP expression of post-selection thymocytes in BH3/Nur77GFP and Bcl-2/Nur77GFP mice, we observed a strikingly similar phenotype. A large population of GFP-high post-selection DP thymocytes was rescued with the expression of either transgene (*Figure 3A*). As in Bim⁻/⁻/Nur77GFP mice, the proportion of post-selection DP thymocytes was consequently increased compared to wild-type mice, reflecting the rescue of these cells from deletion (*Figure 3A*).

To quantify this effect, we defined GFP-high (GFPhi) cells as those expressing GFP at a level greater than the mean fluorescence intensity (M.F.I.) of wild-type post-selection DP cells; this definition allowed for the most consistent gating across independent experiments (*Figure 3A*). The percent increase in GFPhi cells over wild-type is indicative of the proportion of cells rescued from negative selection by transgene expression. Interestingly, a significantly greater increase in the percentage of GFPhi post-selection DP cells was observed in BH3/Nur77GFP vs Bcl-2/Nur77GFP mice, suggesting that the BH3 transgene might indeed have an enhanced ability over wild-type *Bcl2* to rescue high affinity TCR clones from apoptosis (*Figure 3B*). No increase was observed in GFP-intermediate (GFPint) cells, indicating that the GFPhi population encompasses all rescued cells and that BH3 and *Bcl2* transgene expression, similar to Bim-deficiency, only rescues high affinity clones and therefore does not seem to enhance positive selection (*Figure 3A,B*; *Stritesky et al., 2013*). BH3 and *Bcl2* transgene expression also rescued GFPhi CD4 and CD8 SP thymocytes from apoptosis. This likely indicates rescue of high affinity clones from deletion in response to self-antigens presented in the medulla and possibly rescue of DP cells that have differentiated into SP thymocytes. Confirming what has been shown in Nur77GFP mice (*Moran et al., 2011*), GFP expression in BH3 and Bcl-2 Nur77GFP double transgenic mice was also TCR-dependent, as BH3/Nur77GFP and Bcl-2/Nur77GFP SP thymocytes removed from MHC in culture down-regulated GFP similarly to WT/Nur77GFP cells (*Figure 3—figure*

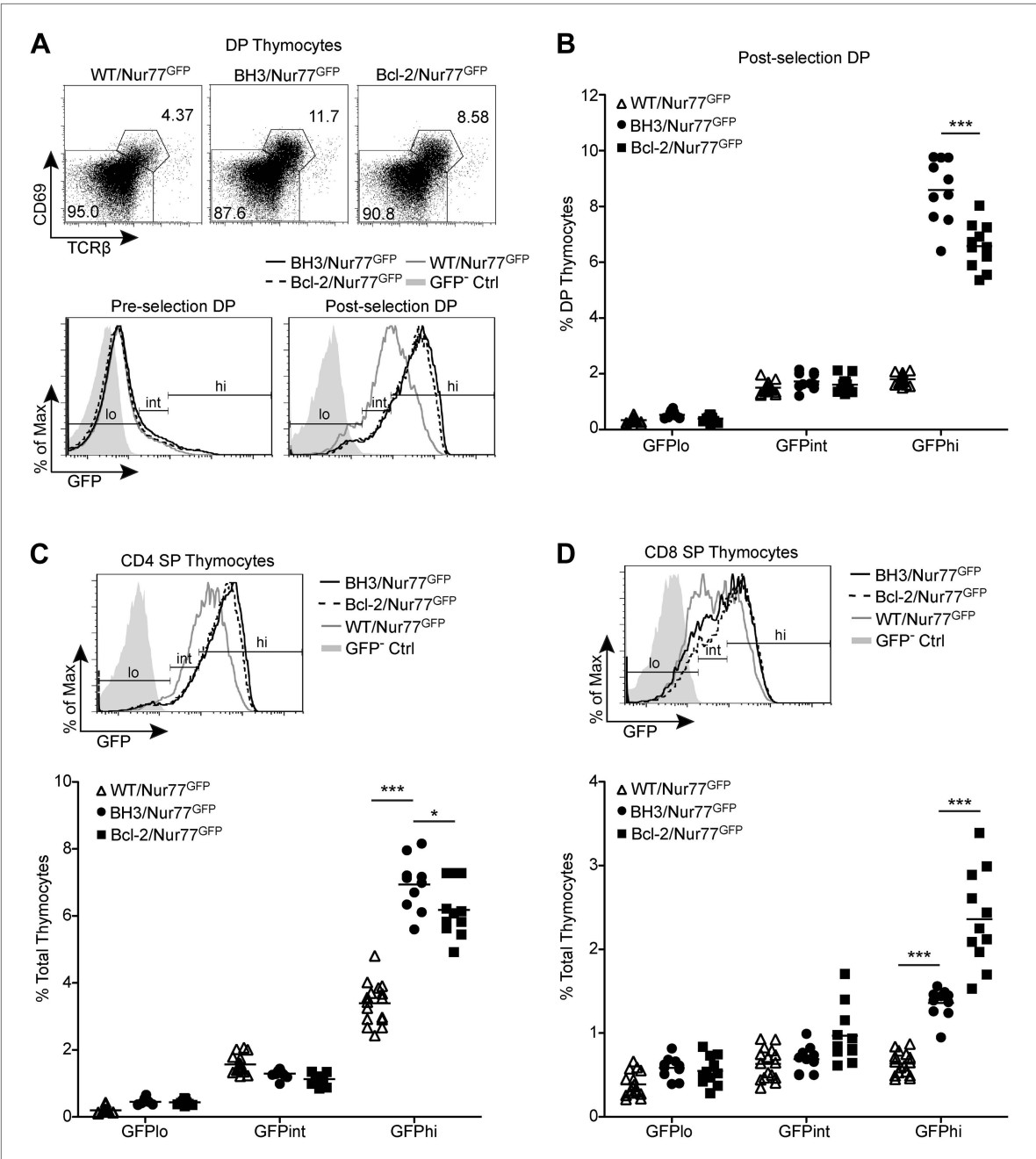

**Figure 3**. BH3 transgene expression rescues high affinity TCR clones in a polyclonal system. (**A**) Flow cytometric analysis of GFP expression in pre-selection (CD69⁻ TCRβ⁻) vs post-selection (CD69⁺ TCRβ⁺) DP thymocytes from Nur77^GFP mice expressing the BH3 and *Bcl2* transgenes. The WT/Nur77^GFP post-selection DP histogram was used to set the GFP low (lo), intermediate (int), and high (hi) gates. The left end of the 'int' gate was set at the base of the GFP⁻ Ctrl histogram and right end was set at the mean GFP fluorescence intensity of the WT/Nur77^GFP histogram. The 'lo' and 'hi' gates were extended from the left and right ends of the 'int' gate, respectively. (**B**) Percentage of DP thymocytes that were post-selection and GFP lo, int and hi as defined in (**A**). (**C** and **D**) Flow cytometric analysis of GFP expression in CD4 SP (**C**) and CD8 SP (**D**) with the gates from post-selection DP cells in A applied. The GFP lo, int and hi percentages were quantified. All mice were 6-weeks-old and WT/Nur77^GFP and GFP⁻ Ctrl mice were littermates to the BH3/Nur77^GFP and Bcl-2/Nur77^GFP mice. Data were pooled from 10 independent experiments.

The following figure supplement is available for figure 3:

**Figure supplement 1**. Nur77 GFP expression in SP thymocytes is regulated by TCR signaling.

*supplement 1*). A greater percentage of GFPhi CD4 SP cells were rescued in BH3 Tg vs Bcl-2 Tg mice (*Figure 3C*). Consistent with the BH3 transgene being downregulated in CD8 SP cells, BH3 expression rescued a smaller percentage of GFPhi CD8 SP thymocytes compared to CD4 SP (*Figure 3D vs 3C*). Collectively, these data demonstrate that Bcl-2 over-expression causes a significant defect in thymocyte negative selection even in a normal, polyclonal system and in this context, the BH3 mutant protein rescues significantly more cells from death than wild-type Bcl-2.

## Defective superantigen negative selection in BH3 Tg mice

Endogenous superantigen negative selection is another polyclonal model of clonal deletion whereby specific Vβ TCR-expressing T cells are deleted through interaction of their specific Vβ chain with endogenously-expressed superantigen bound to class II MHC (usually I–E) on antigen presenting cells (*Scherer et al., 1993*). Whether Bcl-2 over-expression can inhibit negative selection against endogenous superantigens is controversial. In one study, *Bcl2* transgene expression did not block superantigen negative selection (*Sentman et al., 1991*) while another group clearly saw a rescue (*Siegel et al., 1992*). Yet another study reported that cells were rescued from superantigen-mediated deletion in the *Bcl2* transgenic thymus, but not in the transgenic lymph nodes (*Strasser et al., 1991*). To assess endogenous superantigen negative selection in BH3 Tg mice, we crossed them to the class II MHC I-E expressing CBA/J strain of mice. CBA/J mice express superantigens that delete T cells expressing Vβ3, 5, 6, 7, 8.1, 9 and 11, but not Vβ8.2 (*Scherer et al., 1993*). We examined the proportions of Vβ5, Vβ6, Vβ8.1/8.2 and Vβ11 expressing DP and SP thymocytes in BH3 Tg X CBA/J mice and their wild-type littermate controls as well as in Bcl-2 Tg X CBA/J mice. As shown in *Figure 4*, the proportions of Vβ5, Vβ6 and Vβ11, but not Vβ8.1/8.2, expressing cells decreased between the DP and SP stage in non-transgenic mice, indicative of superantigen negative selection. In both BH3 and Bcl-2 Tg mice, however, increased proportions of Vβ5, Vβ6 and Vβ11 expressing SP thymocytes were observed, suggesting that expression of either transgene can block superantigen-mediated deletion. The BH3 transgene rescued CD8 SP thymocytes less efficiently than the wild-type *Bcl2* transgene, reflective of the relative levels of transgenic protein in the CD8 compartment. CD4 SP thymocytes were rescued to a similar extent by both transgenes, however modestly better by the BH3 transgene for Vβ5[+] and Vβ6[+] cells. Interestingly, we observed a significant increase in Vβ5[+] and Vβ6[+] (but not Vβ11[+]) DP thymocytes in the BH3 Tg mice that was not seen in the Bcl-2 Tg mice. The pattern of rescue in this model parallels that observed in the Nur77[GFP] polyclonal system. Together, these data are consistent with the notion that overexpression of both wild type and BH3 mutant *Bcl2* transgenes leads to an enhanced rescue of thymocytes from negative selection, with the BH3 mutant providing a more complete rescue.

## Alternative tolerizing mechanisms are increased in BH3 Tg mice

Our results demonstrate that T cell-specific over-expression of wild-type Bcl-2 clearly results in a rescue of cells receiving strong TCR signals, which would normally undergo apoptosis, yet aged Bcl-2 Tg mice were not reported to develop autoimmune pathology (*Linette et al., 1995*). Hence, we sought to investigate the fate of the rescued, high affinity T cell clones in the Bcl-2 Tg and BH3 Tg mice. We observed that the GFP mean fluorescence intensity was higher in lymph node T cells from both BH3/Nur77[GFP] and Bcl-2/Nur77[GFP] mice (*Figure 5A*). GFP expression was also dependent on TCR signaling, as removal of BH3/Nur77[GFP] and Bcl-2/Nur77[GFP] mature T cells from MHC in culture resulted in downregulation of GFP at a rate similar to that observed for WT/Nur77[GFP] cells (*Figure 5—figure supplement 1*). Hence, the increased GFP expression might be leftover from strong TCR signals received in the thymus and/or due to tonic TCR stimulation in the periphery. Interestingly, GFP levels were higher in Bcl-2 Tg vs BH3 Tg T cells, even for CD4[+] T cells. The reason for this is unclear, but it may be due to distinct T cell repertoires in these two lines of mice that lead to differences in T cell signaling in the periphery. T$_{reg}$ cells have self-reactive TCRs and express a high level of GFP, however T$_{reg}$ cells did not account for the majority of the GFP-high CD4 T cells (data not shown). This indicates that conventional T cells with higher than normal TCR affinity are permitted to migrate into the periphery. Since presumably autoreactive T cells are found in the periphery, mechanisms must exist to keep these cells in check. One mechanism might be the production of more T$_{reg}$ cells to mediate suppression of the autoreactive cells (*Stritesky et al., 2012*). Interestingly, we observed that GFP-high CD4 SP thymocytes rescued from negative selection express GFP at a very similar level to T$_{reg}$ cells (*Figure 5B*). Since high TCR affinity contributes to T$_{reg}$ development (*Stritesky et al., 2012*), we hypothesized that more thymocytes might be directed down this pathway in BH3 and Bcl-2 Tg mice. Indeed, we found greater Foxp3[+] T$_{reg}$ cell proportions and numbers in the both the thymus and lymph nodes of BH3

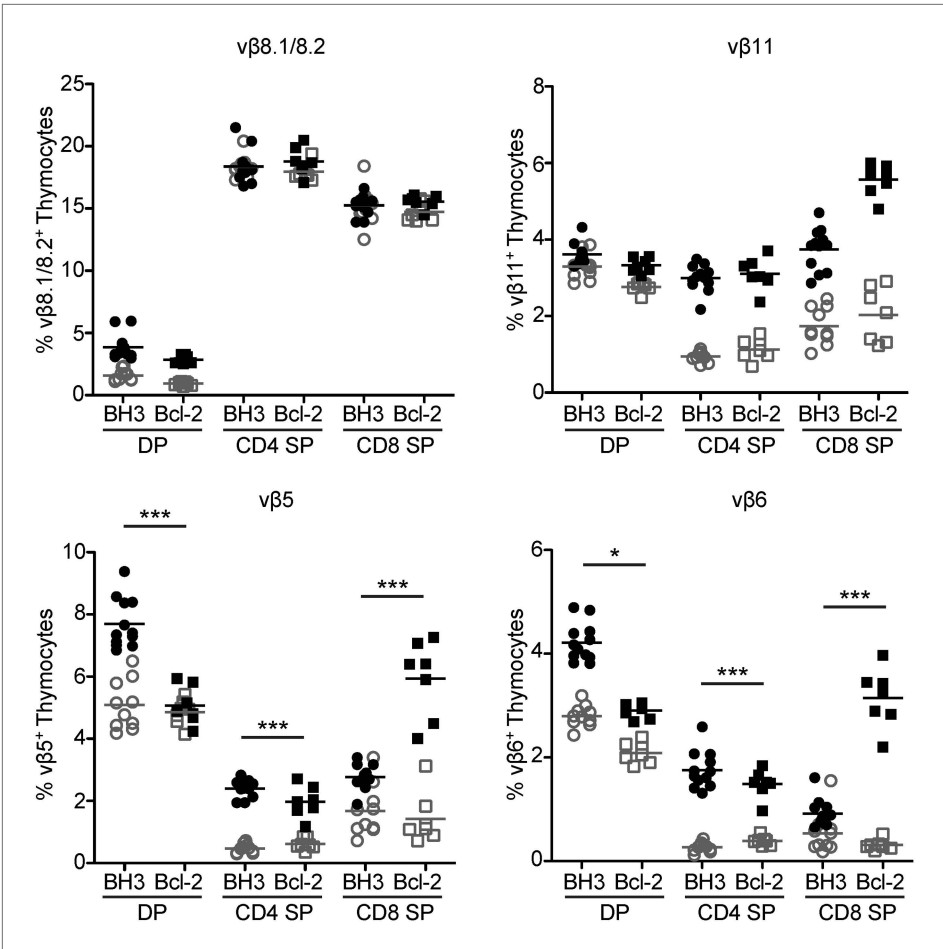

**Figure 4**. Superantigen negative selection is inhibited in BH3 Tg mice. Flow cytometric analysis of TCR vβ chain expression in DP, CD4 SP and CD8 SP thymocyte subsets from BH3 Tg and Bcl-2 Tg mice (closed shapes, black) compared to littermate wild-type controls (open shapes, grey) on a C57BL/6 X CBA/J genetic background. All mice were 8-weeks-old.

and Bcl-2 Tg mice (*Figure 5C*). BH3 Tg mice had a more significant increase in peripheral $T_{reg}$ numbers compared to Bcl-2 Tg mice (3-fold vs 2-fold), mirroring the higher mature CD4 T cell numbers in these mice. Most of the increase in total $T_{reg}$ number was caused by an increase in CD25$^-$, and not CD25$^+$ cells. CD25$^-$ $T_{reg}$ cells are thought to be derived from diverted, potentially autoreactive thymocytes and are functional, but have reduced suppressive capacity compared to CD25$^+$ $T_{reg}$ cells (*Zhan et al., 2011*). Despite the increase in total $T_{reg}$ number in BH3 and Bcl-2 Tg mice, the ratio of CD25$^+$ $T_{reg}$ cells to total CD4 T cells was similar to wild-type mice (*Figure 5C*). Thus, some of the rescued high affinity clones appear to be diverted down the CD25$^-$ $T_{reg}$ pathway, but not the CD25$^+$ $T_{reg}$ pathway. Intriguingly, consistent with this, we observed that CD25$^-$ $T_{reg}$ cells express higher levels of GFP than CD25$^+$ $T_{reg}$ cells, at a similar level to the highest GFP-expressing BH3 Tg clones (*Figure 5D*). Another mechanism employed to maintain peripheral tolerance is the induction of T cell anergy upon antigen encounter. In addition to increased $T_{reg}$ cells, we also observed a higher proportion of previously activated, anergic CD4 T cells in BH3 and Bcl-2 Tg mice marked by upregulation of the markers FR4 and CD73 in CD44hi T cells (*Figure 5E–F*; *Martinez et al., 2012*). Therefore, increased diversion to the $T_{reg}$ lineage/$T_{reg}$ suppression and CD4 T cell anergy might cooperate to suppress the emergence of autoimmunity in these mice.

## BH3 Tg mice have a shortened lifespan and lymphoid hyperplasia

Despite similarly engaging alternative mechanisms to promote tolerance, BH3 Tg mice, unlike Bcl-2 Tg mice, have a considerably shortened life-span, with a median length of survival of about 50 weeks for

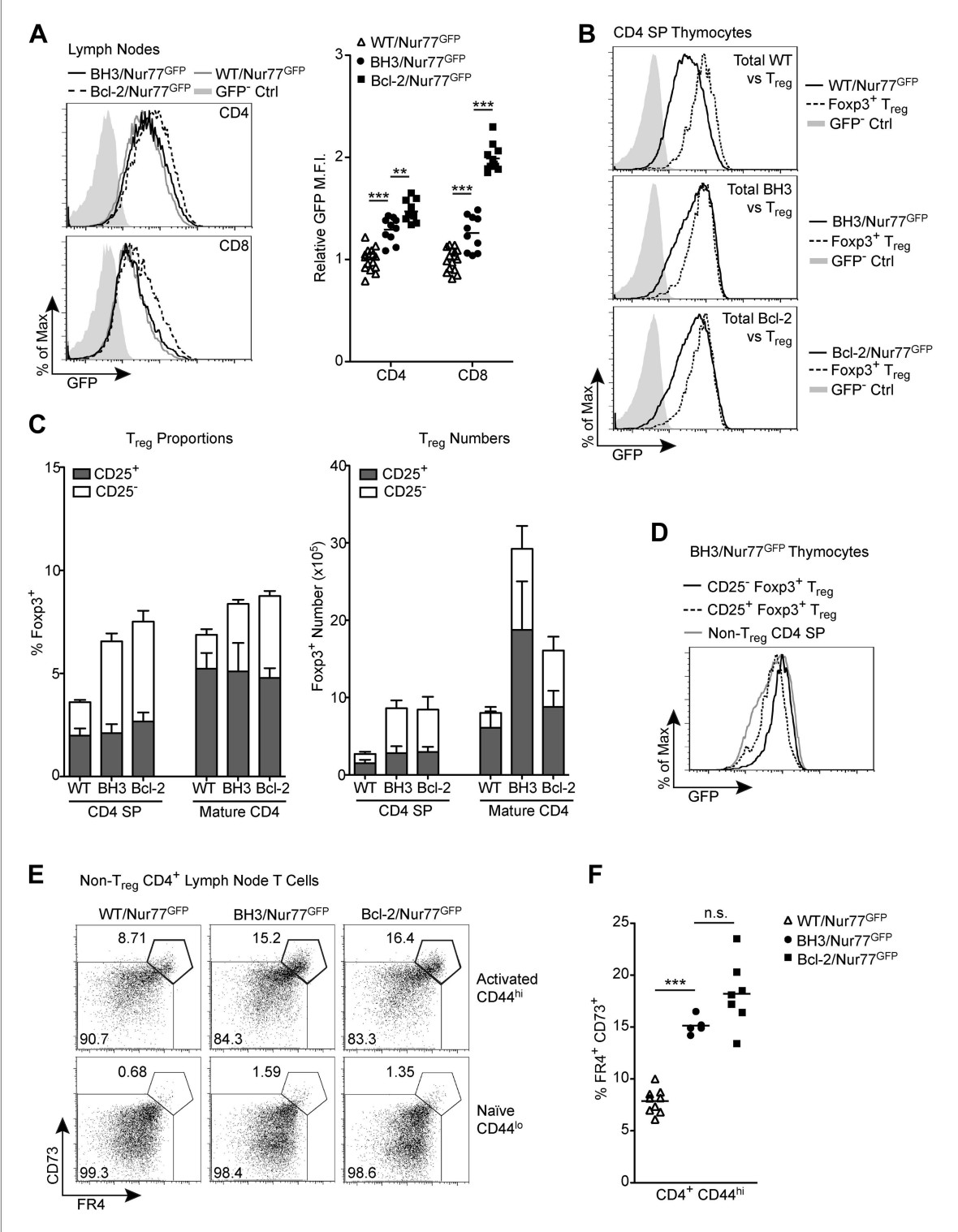

**Figure 5**. Alternative tolerance mechanisms in BH3 Tg mice. (**A**) Flow cytometric analysis of GFP expression by mature CD4 and CD8 lymph node T cells. Quantification indicates the fold change in GFP mean fluorescence intensity (M.F.I.) relative to WT/Nur77GFP CD4 and CD8 samples. GFP M.F.I. was normalized between experiments by subtracting the M.F.I. of a GFP− control. n ≥ 10 per genotype. (**B**) Comparison of GFP expression by total CD4 SP thymocytes vs Foxp3+ CD4 SP Treg cells. (**C**) Quantification of flow cytometric analysis of CD25+ vs CD25− Treg cells (CD4+ Foxp3+) in the thymus and lymph nodes. n ≥ 7 per genotype. (**D**) Comparison of GFP expression levels in CD25+ vs CD25− Treg cells from BH3 Tg mice. (**E**) Flow cytometric analysis of non-Treg, anergic CD4 T cells in the lymph nodes. Treg cells were gated out by high expression of GITR and CD25. Anergic cells, indicated by the bolded

*Figure 5. Continued on next page*

Figure 5. Continued

gates, express high levels of CD44, FR4 and CD73. (**F**) Quantification of anergic CD4 T cells gated as described in (**E**). Mice were 6-weeks-old in (**A**–**D**) and 7-weeks-old in (**E**–**F**). All data are representative of or compiled from at least three independent experiments.

The following figure supplement is available for figure 5:

**Figure supplement 1**. Nur77 GFP expression in mature T cells is maintained by TCR signaling.

both BH3 A and BH3 B lines (**Figure 6A**). Most mice were sacrificed due to general deterioration of health, especially dramatic weight-loss, but some mice of the BH3 A line were sacrificed due to a severe inflammatory skin condition. No Bcl-2 Tg mice had to be sacrificed over a 60-week period. Upon examination of a large cohort of mice between 45 and 60 weeks-of-age, we found that all BH3 Tg mice displayed severe splenomegaly and lymphadenopathy and had significantly increased numbers of both T cells and B cells, especially in the mesenteric lymph nodes (MLN) and spleen (**Figure 6B,C**). In comparison, Bcl-2 Tg lymphoid organs were only mildly enlarged, but a significant increase in T cell numbers was observed. We also observed a dramatic increase in $CD3^-$ $B220^-$ Myeloid/other cells in the MLN and spleen of BH3 Tg, but not Bcl-2 Tg mice (**Figure 6D**). A large proportion of these cells were $Ly6C^{lo}$ $CD11b^+$ neutrophils, indicative of increased inflammation in these mice (**Figure 6E**).

## BH3 Tg mice accumulate activated, autoreactive T cells

CD4 and CD8 T cells in aged BH3 Tg mice were highly activated, as indicated by increased CD44 expression and decreased CD62L expression, reminiscent of the phenotype of the disease-promoting T cells in AIRE-deficient mice [**Figure 7A**; (**Anderson et al., 2002**)]. We also observed a significant increase in CD44hi/CD62Lhi central memory CD8 T cells. An increase in activated T cells was also sometimes observed in Bcl-2 Tg mice, but the phenotype was greatly attenuated. While we did not observe an increase in activated T cells in young mice (data not shown), starting as early as 20 weeks-of-age, there was significant accumulation of activated T cells in the peripheral blood of BH3 Tg mice (**Figure 7B**). This increase became very dramatic and nearly ubiquitous in 40 to 50-week-old mice. In comparison, only a couple Bcl-2 Tg mice at 40 to 50 weeks were found to have increased activated T cells. To directly test the possible autoreactivity of BH3 Tg T cells, we stimulated T cells with syngeneic or allogeneic irradiated splenocytes and measured T cell proliferation in a mixed lymphocyte reaction (MLR). All T cells were able to proliferate in response to allogeneic stimulators, although CD4 T cells proliferated less than CD8 T cells and the presence of Bcl-2 further reduced proliferation (**Figure 7C**). This datum is consistent with the known inhibitory effect of Bcl-2 on cell cycle progression (**Linette et al., 1996**; **Cheng et al., 2004**). When presented with syngeneic stimulators, little proliferation was observed for all the Tg and wild-type CD4 T cells, even with $T_{reg}$ depletion, most likely due to a combination of the lackluster ability of CD4 T cells to proliferate in this system and the inhibitory effect of Bcl-2 on cell cycle progression. In contrast, $CD8^+$ BH3 Tg T cells exhibited significant proliferation in response to syngeneic stimulators, suggesting that the BH3 Tg mice contain autoreactive T cells (**Figure 7C**). Despite our inability to detect autoreactive CD4 T cells in BH3 Tg mice by MLR, we found that the activated CD4 T cells in aged mice appear to have increased effector function, as splenic CD4 T cells from 50-week-old mice secrete increased amounts of IFNγ (**Figure 8A**). Only IFNγ and not IL-17 or IL-4 was increased, suggesting that the activated T cells are Th1 polarized (**Figure 8A**). Autoimmunity manifested as significant infiltration of lymphocytes into the liver, lung and kidney of BH3, but not Bcl-2 Tg mice (**Figure 8B**). In addition, autoantibodies against multiple non-lymphoid tissues (pancreas, eye, lung, liver, kidney, stomach) were detected by immunoblot of whole tissue extracts with sera from 40 to 50-week-old BH3 Tg mice (**Figure 8C**). Interestingly, autoantibody binding was detected for all the BH3 Tg mice examined and most if not all the mice had autoantibodies against multiple tissues. Common banding patterns were observed amongst the BH3 Tg mice within a given tissue suggesting common autoantibody targets. Occasionally, autoantibodies were also found in some Bcl-2 Tg mice, but to a much lesser extent. An occasional band was sometimes observed in one of the wild-type controls, but these bands did not match the patterns observed in the BH3 Tg mice. A few BH3 Tg mice also exhibited a significant increase in anti-nuclear antibodies (ANA) (**Figure 8D**). However, for the majority of the BH3 Tg mice, the levels were not appreciably higher than wild-type or Bcl-2 Tg mice, perhaps consistent with the autoimmune pathology in these mice being more predominantly T cell-rather than B cell-driven. In a few mice with high levels of ANA, significant disruption of kidney

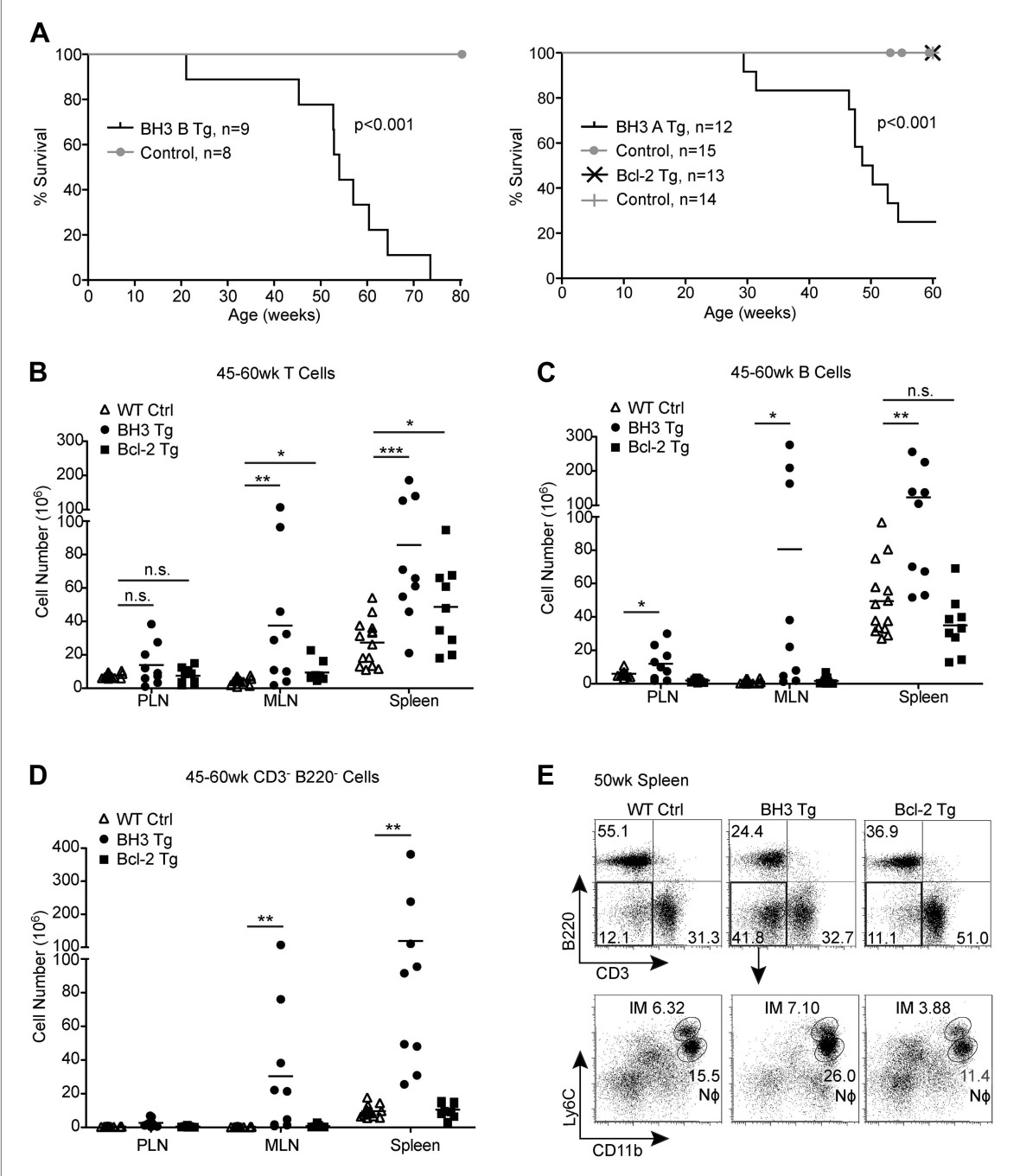

**Figure 6**. BH3 Tg mice have a shortened lifespan and exhibit lymphoid hyperplasia. (**A**) Kaplan-Meier curves for BH3 Tg lines A and B and Bcl-2 Tg mice compared to littermate WT controls. Statistical significance was calculated using the Log-rank test. (**B–D**) Quantification of CD3+ T cells (**B**), B220+ B cells (**C**) and CD3− B220− Myeloid/other cells (**D**) in the peripheral lymph nodes (PLN), mesenteric lymph nodes (MLN) and spleen of 45 to 60-week-old mice. WT controls were littermates to BH3 Tg and Bcl-2 Tg mice. (**E**) Flow cytometric analysis of CD11b and Ly6C expression on CD3− B220− splenocytes from 50-week-old mice. Ly6Chi CD11b+ inflammatory monocytes (IM) and Ly6Clo CD11b+ neutrophils (Nφ, also Ly6G+) are gated. Plots are representative of at least five mice per genotype.

morphology was observed, reminiscent of glomerulonephritis (*Figure 8B*). Aged BH3 Tg mice also had significantly increased numbers of both CD25+ and CD25− $T_{reg}$ cells in their lymphoid organs, making up approximately 50% of all CD4 T cells (*Figure 8E–F*). This dramatic increase in $T_{reg}$ cells was still apparently insufficient to inhibit autoimmunity in these mice.

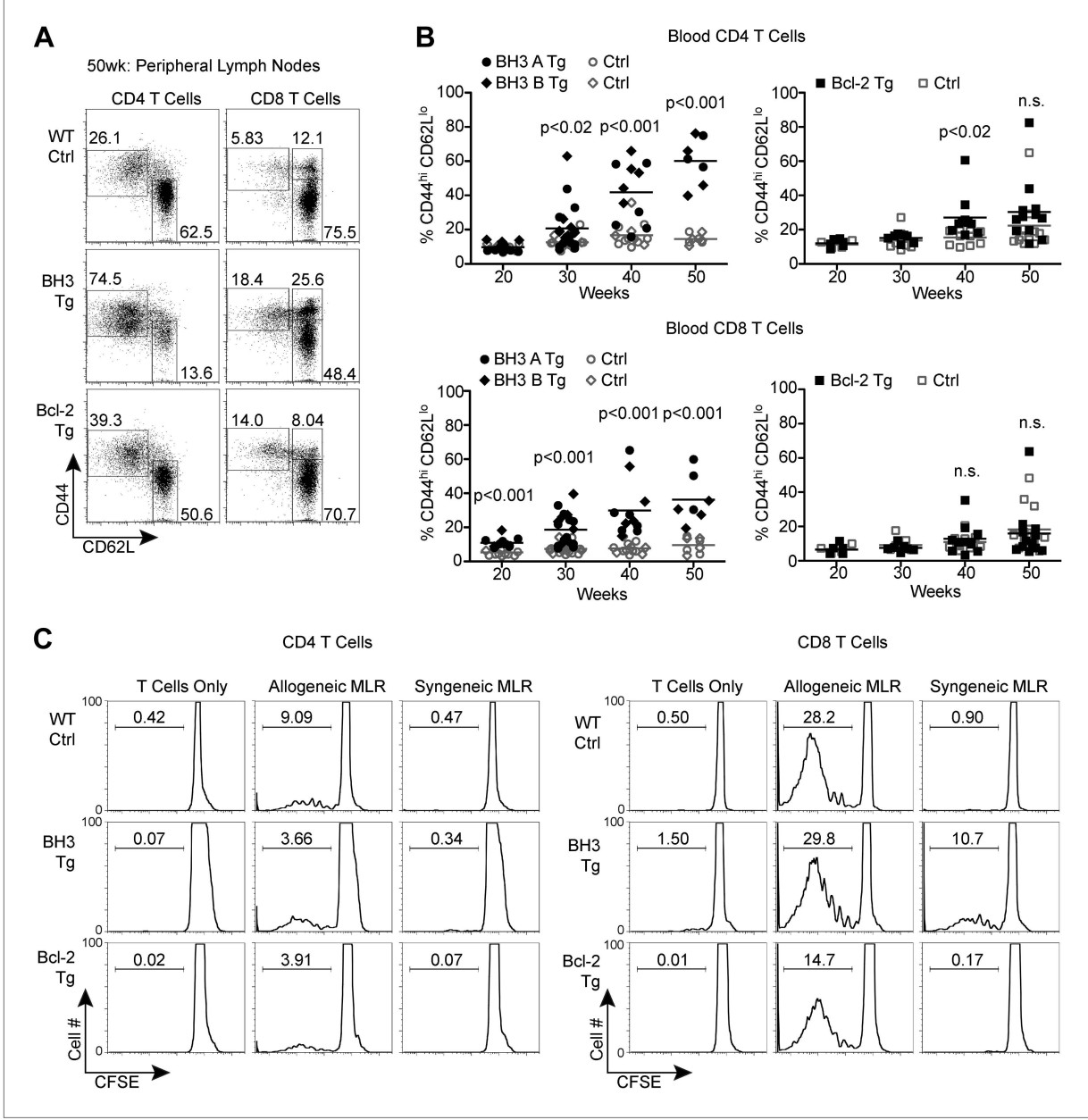

**Figure 7**. BH3 Tg mice accumulate activated, autoreactive T cells. (**A**) Flow cytometric analysis of activated CD4 and CD8 peripheral lymph node T cells by CD44 and CD62L expression in 50-week-old mice. For CD4 T cells, gates indicate activated (CD44$^{hi}$ CD62L$^{lo}$) and naïve (CD44$^{lo}$ CD62L$^{hi}$) populations. For CD8 T cells, activated effector memory (CD44$^{hi}$ CD62L$^{lo}$) and activated central memory (CD44$^{hi}$ CD62L$^{hi}$) vs naïve (CD44$^{lo}$ CD62L$^{hi}$) populations are gated. (**B**) Compiled analyses of the percentage of activated T cells (CD44$^{hi}$ CD62L$^{lo}$) in the peripheral blood of aged mice. Controls (Ctrl) are littermates to the Tg mice as denoted by data point shape. (**C**) Mixed lymphocyte reaction: purified T cells from 20-week-old mice were cultured with allogeneic (Balb/c) or syngeneic (C57BL/6) irradiated splenocytes at a 1:1 ratio. Proliferation was measured by CFSE dilution over 4 days. Histograms are representative of triplicate samples and five independent experiments, two with T$_{reg}$ cell-depletion (shown) and three without. Gates indicate the percentage of total cells that have undergone at least one round of proliferation.

## Discussion

Bcl-2 family proteins and the Nur77 family of orphan steroid receptors are critical mediators of thymocyte apoptosis during negative selection. However, whether thymocyte deletion is necessary to maintain immunological self-tolerance remains under debate. Here, we find that T cell-specific over-expression of a *Bcl2* BH3 mutant transgene results in T cell-driven multi-organ autoimmunity, strongly

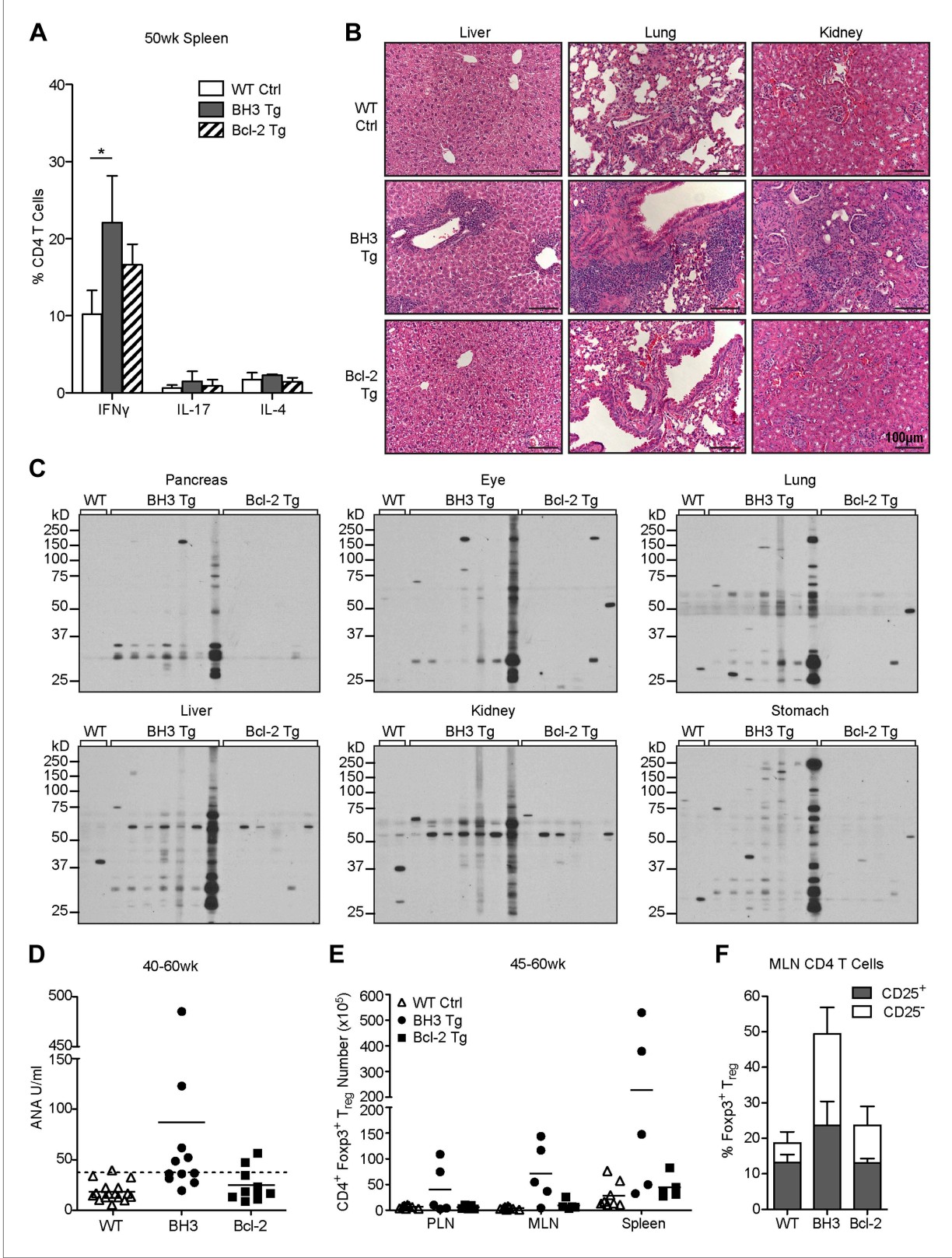

**Figure 8**. T cell autoimmune pathology is apparent in BH3 Tg mice. (**A**) Quantification of flow cytometric analysis of T helper cell cytokine expression. Splenic CD4 T cells from 50-week-old mice were stimulated with PMA and Ionomycin for 4 hr in the presence of brefeldin A to allow cytokine accumulation. n ≥ 3 mice per genotype. (**B**) Hematoxylin and Eosin staining for lymphocyte infiltrates in liver, lung and kidney sections from 50-week-old mice.
*Figure 8. Continued on next page*

*Figure 8. Continued*

Images were captured at 20x magnification. Data are representative of at least three mice per genotype. (**C**) Sera immunoblots for detection of autoantibodies. Whole tissue extracts were probed with sera from 40 to 50-week-old mice. Each lane corresponds to sera from an individual mouse: WT n = 2, BH3 Tg n = 7, Bcl-2 Tg n = 6. (**D**) Serum anti-nuclear antibody (ANA) quantification by ELISA. Dashed line represents two standard deviations above the WT mean. Samples above this line are considered positive for ANA. (**E**) Quantification of $CD4^+$ $Foxp3^+$ $T_{reg}$ cells in the lymphoid organs of aged mice. (**F**) Percentage of CD4 T cells that are $T_{reg}$ cells ($CD25^+$ vs $CD25^-$) in the mesenteric lymph nodes of 45 to 60-week-old mice. n ≥ 5 mice per genotype.

suggesting that a defect in thymic deletion is indeed sufficient to cause a breakdown in tolerance. Consistent with previous observations, we found that T cell-specific over-expression of wild-type Bcl-2 did not cause significant autoimmune pathology (*Linette et al., 1995*). However, our results show that Bcl-2 Tg mice may develop autoimmunity at low penetrance, as some mice exhibited a modest accumulation of activated T cells and autoantibodies. In contrast, expression of a *Bcl2* BH3 mutant transgene significantly increased the severity of the disease and resulted in greater than 70% lethality by one year-of-age. A significant accumulation of activated T cells was observed starting at 20 to 30 weeks-of-age and ultimately resulted in massive lymphoid hyperplasia, inflammation, lymphocyte infiltration into non-lymphoid organs and organ-targeted auto-antibody accumulation. This phenotype of low ANA antibody titers, activated T cells and organ-targeted autoantibodies is suggestive of a primarily T cell-mediated disease.

Our data in the Nur77^GFP and superantigen mouse models indicate that the Bcl-2 BH3 mutant is better able to rescue autoreactive thymocytes from negative selection than wild-type Bcl-2. Abolishing the Nur77-dependent pro-apoptotic function of Bcl-2 by BH3 mutation might therefore enhance the anti-apoptotic effects of Bcl-2 over-expression. Alternatively, it is also possible that slightly higher BH3 transgene expression levels in DP and CD4 SP cells could be responsible for this phenotype, although the levels of transgene expression do not necessarily correlate with GFP expression as demonstrated by the higher GFP M.F.I. in $CD4^+$ T cells of Bcl-2 over BH3 Tg mice (*Figure 5A*). Cause set aside, a greater defect in negative selection could allow a threshold to be crossed in which other mechanisms of maintaining tolerance are no longer sufficient to prevent disease. This would be consistent with the recent study comparing the severity of disease in $Bim^{-/-}$ vs $Bim^{-/-}Bbc3^{-/-}$ mice. $Bim^{-/-}$ mice on the C57BL/6 background were reported to have attenuated autoimmunity and 80% of them still survive at 70 weeks-of-age (*Bouillet et al., 2001*; *Labi et al., 2014*). Additional loss of Puma (*Bbc3*), however, enhanced the thymic deletion defect of $Bim^{-/-}$ mice and resulted in more severe, T cell-driven multi-organ autoimmune pathology (*Gray et al., 2012*). It remains to be seen if loss of Bim and Puma in T cells alone would result in the same phenotype. Our observations in a T cell-specific model provide strong evidence that a severe thymic clonal deletion defect is sufficient to cause autoimmune pathology. Arguably, a defect in peripheral T cell deletion due to increased T cell survival may also contribute to the BH3 Tg phenotype. Although we cannot completely rule out this possibility, the fact that Bcl-2 Tg mice do not develop autoimmunity would suggest a minimal role for defective peripheral tolerance in this model.

Despite the large disparity in autoimmune phenotype, we found that both BH3 and Bcl-2 Tg thymocytes were significantly protected from apoptosis in TCR transgenic models of negative selection. Previously, Bcl-2 over-expression was directly compared with Bim-deficiency in the HY model and Bcl-2 over-expression fell significantly short of Bim-deficiency in rescuing thymocytes from deletion (*Bouillet et al., 2002*). However, in our thymic slice assay of HY TCR-mediated deletion, we find that Bcl-2 over-expression almost completely blocks thymocyte death. Similarly, over-expression of Bcl-2 in F5 TCR transgenic mice dramatically blocked thymocyte deletion by thymic slice assay and in vivo peptide injection. One possible explanation for the discrepancy between our results and previously published observations might be the negative selection models used. Apoptosis of HY TCR transgenic mice is notoriously difficult to block due to accelerated T cell development and early expression of the transgenic TCR (*Baldwin et al., 2005*). Indeed, using thymic organ culture, another group has previously reported a significant inhibition of negative selection by Bcl-2 in the F5 model (*Williams et al., 1998*). Regardless, however, we have demonstrated that Bcl-2 over-expression has a potent anti-apoptotic effect on negative selection in a normal, polyclonal system. Mirroring observations in $Bim^{-/-}$/Nur77^GFP mice (*Stritesky et al., 2013*), we observed a striking increase in GFP-high post-selection thymocytes

in Bcl-2/Nur77^GFP mice, indicating a significant rescue of high affinity TCR clones from deletion. Here, over-expression of the BH3 mutant transgene resulted in substantially greater rescue of GFP-high DP and CD4 SP thymocytes compared to over-expression of wild-type *Bcl2*. BH3 transgene expression also enhanced rescue of vβ5 and vβ6 DP and CD4 SP thymocytes from deletion in a polyclonal superantigen model of negative selection. Increased rescue of these cells in the thymus might lead to the development of a more self-reactive T cell repertoire, accelerating the expansion of activated T cells in BH3 Tg mice.

Differential expression of the BH3 vs wild-type *Bcl2* transgenes in the CD8 T cell compartment might also contribute to the development of autoimmunity in BH3 Tg, but not Bcl-2 Tg mice. *Cd4*-driven transgene expression in the BH3 transgenic mice preferentially enhances CD4 T cell seeding of the peripheral lymphoid organs. BH3 Tg mice have about a 3:1 ratio of CD4 to CD8 T cells in the lymph nodes compared to a 2:1 ratio in wild-type mice. In Bcl-2 Tg mice, the proportions are skewed in the opposite direction, resulting in a 1:1 ratio. This is likely due to a combination of transgene-biased rescue of cells from negative selection and increased survival of transgene-expressing T cells. BH3 Tg mice are therefore expected to have increased numbers of autoreactive CD4 T cells, which as key mediators of autoimmunity, could be responsible for the acceleration of disease in these mice (*Palmer and Weaver, 2010*). Alternatively, as Bcl-2 over-expression reduces the ability of T cells to proliferate (*Linette et al., 1996*; *Cheng et al., 2004*), downregulation of the BH3 transgene in CD8 T cells might enhance their functional capacity over that of Bcl-2 Tg CD8 T cells. As CD8 T cells play a well-documented role in T cell-mediated, tissue-specific autoimmune diseases (i.e. multiple sclerosis and diabetes), more functionally competent autoreactive CD8 T cells in the BH3 Tg mice might contribute to the development of disease (*Gravano and Hoyer, 2013*).

In response to defective clonal deletion in the thymus, we observed compensatory induction of alternative tolerizing mechanisms that might prevent earlier onset of autoimmunity in BH3 Tg mice. An increase in the proportion and number of $T_{reg}$ cells was noted in both the thymus and lymph nodes of young mice. Interestingly, CD25^− $T_{reg}$ cells accounted for most of the increase in total $T_{reg}$ numbers and these cells have been reported to have reduced suppressive function (*Zhan et al., 2011*). These CD25^− $T_{reg}$ cells are thought to be derived from diverted, autoreactive cells that were not deleted in the thymus. Consistent with this, CD25^− $T_{reg}$ cells in the thymus expressed GFP at a higher level than CD25^+ $T_{reg}$ cells and at a similar level to the most GFP-high naïve CD4 SP cells in the BH3/Nur77^GFP mice. Hence, diverting a portion of these GFP-high autoreactive cells to this lineage might be the more effective tolerizing strategy. Notably, our observation that CD25^+ $T_{reg}$ proportions were not increased in the BH3 Tg mice suggests that T cells with self-reactive TCRs are not automatically committed to the $T_{reg}$ lineage and that there is likely a saturated niche for CD25^+ $T_{reg}$ development determined by availability of other factors, such as IL-2 and IL-15 (*Stritesky et al., 2012*). In addition to increased $T_{reg}$ cells, we observed a 2-fold increase in the proportion of anergic CD4 T cells in the BH3 and Bcl-2 Tg mice. This was similarly observed in *Bim*^−/− mice (*Stritesky et al., 2013*). Induction of anergy could cooperate with diversion to the $T_{reg}$ lineage/$T_{reg}$ suppression to inhibit T cell-mediated promotion of autoimmunity and prevent early onset of disease. However, differential induction of these tolerizing mechanisms was not observed in BH3 Tg vs Bcl-2 Tg mice and therefore cannot account for the eventual development of autoimmunity in BH3 Tg, but not Bcl-2 Tg mice. Interestingly, in aged BH3 Tg mice, $T_{reg}$ cells made up almost 50% of the CD4 T cell pool in mice displaying autoimmune pathology. However, this protective response was still apparently insufficient to suppress autoimmunity.

Finally, the enhanced ability of the *Bcl2* BH3 mutant transgene to rescue autoreactive thymocytes from apoptosis compared to wild-type *Bcl2* could suggest that the Bcl-2 BH3 domain might indeed have some pro-apoptotic function. This finding may provide in vivo support for the notion that Nur77 promotes thymocyte apoptosis through conversion of Bcl-2 to a pro-apoptotic effector. However, this still needs to be further confirmed in a knock-in model. Additionally, analysis of a Bcl-2 protein that cannot interact with Nur77 would also be required to confirm that Bcl-2 pro-apoptotic function is Nur77-dependent. Intriguingly, it has previously been suggested that Nur77 and Bim might function in the same pathway, as Bim-deficiency was unable to synergize with Nur77-deficiency to inhibit thymocyte apoptosis in the OTII TCR transgenic model of negative selection (*Fassett et al., 2012*). If Bcl-2 conversion is indeed critical for Nur77-mediated thymocyte apoptosis, this would support the idea that Nur77 and Bcl-2 family members come together at the mitochondria to cooperatively regulate cell death during negative selection.

## Materials and methods

### Mice

The mutant allele for the *Bcl2* BH3 mutant transgenic mice (BH3 Tg) was generated by QuikChange site-directed mutagenesis (Agilent Technologies, Santa Clara, CA) from a pCI-human *Bcl2* vector using the following primers: BH3 forward CTGACCCTCCGCCAGGCCGCGGCCGCCTTCTCCCGCCGCTACCGC, BH3 reverse GCGGTAGCGGCGGGAGAAGGCGGCCGCGGCCTGGCGGAGGGTCAG. The nucleotide sequence of amino acids 101-103 was changed from GGCGACGAC to GCGGCCGCC, creating a restriction enzyme site change from BglI to NotI. The mutant transgene was then cloned into the XhoI site of the pTG4 construct (*Adlam and Siu, 2003*; *Xue et al., 2010*). The pTG4 construct was then modified to include the *Cd4* locus control region (LCR). PCR amplification of the 1 kb LCR sequence from C57BL/6 genomic DNA was performed with the following primers: LCR forward GACATCGATAGCTAGCACACGCCGGTAAGCCCATTCCCCACGC, LCR reverse GACATCGATGCGGTACCGATCCCAACCAAACTGCGGCCCTTTCA. The LCR was added to the 5′ end of the *Cd4* silencer at a ClaI restriction site. The transgenic mice were generated on the C57BL/6 background using standard procedures. Founders were identified by PCR genotyping with the following primers: HGH forward GACACAAACTCACACAACGATGACGC, HGH reverse ATGCCTGGAATCCCAACAACTCGG. The presence of the BH3 GDD to AAA mutation in the incorporated transgene was confirmed by PCR amplification of the transgene from genomic DNA and successful digestion with NotI, but not BglI enzyme. The BH3 Tg mice were compared with the T cell-specific wild-type *Bcl2* transgenic line *Lck^Pr*-*Bcl2* (*Sentman et al., 1991*), which have been back-crossed to C57BL/6 for more than 10 generations, referred to here as Bcl-2 Tg. The BH3 and Bcl-2 Tg mice were crossed with the following transgenic lines: Nur77^GFP (*Moran et al., 2011*), F5 TCR (*Mamalaki et al., 1993*), and HY TCR (*Teh et al., 1990*). All mice were on the C57BL/6 background. All experimental protocols involving animals were approved by the UC Berkeley Animal Care and Use Committee.

### Flow cytometry

Lymphoid organs were dissociated through 40 μm cell strainers to obtain a single cell suspension and a red blood cell lysis was performed. Cells were surface stained with fluorochrome-conjugated antibodies (eBioscience, San Diego, CA, BD Biosciences, San Jose, CA and Tonbo Biosciences, San Diego, CA) in 1% fetal bovine serum in PBS. For Annexin V staining, after surface staining, cells were stained with FITC-conjugated Annexin V (BD Biosciences) in 1x Annexin V binding buffer (10x: 0.1M HEPES, pH 7.4; 1.4M NaCl; 25 mM $CaCl_2$). For intracellular staining, prior to surface staining, cells were stained with Tonbo Ghost Dye (Tonbo Biosciences) to label dead cells, then fixed and permeabilized with a Cytofix/Cytoperm Fixation/Permeabilization Kit (BD Biosciences). For Bcl-2 intracellular staining, cells were then stained with FITC-conjugated anti-human Bcl-2 (Clone 124, Dako, Carpinteria, CA) or a FITC-conjugated mouse IgG1 isotype control (Dako). For Cleaved Caspase 3 staining, cells were incubated with 5% Normal Donkey Serum and anti-CD16/32 Fc block (Cone 2.4G2, UCSF Hybridoma Facility, San Francisco, CA) for 15 min then stained with anti-Cleaved Caspase 3 (Asp175, Cell Signaling, Danvers, MA) or a rabbit IgG isotype control (Santa Cruz Biotechnology, Dallas, TX) for 45 min. For detection of the cleaved caspase-3, a PE donkey anti-rabbit secondary antibody (Jackson ImmunoResearch, West Grove, PA) was added for 20 min. For intracellular cytokine staining, cells were incubated with FITC-IFNγ (Clone XMG1.2), APC-IL-17 (Clone eBio17B7) and PE-IL-4 (Clone 11B11) or similarly conjugated isotype controls, all from eBioscience, for 30 min. Intracellular staining for Foxp3 was performed with an eBioscience PE anti-Foxp3 staining kit (Clone FJK-16 s) following dead cell and surface staining. All samples were analyzed on the BD Biosciences LSR Fortessa or LSR II.

### Western Blotting

Lymphocyte single cell suspensions were pelleted, washed with PBS, and lysed in 1% NP-40 lysis buffer (150 nM NaCl, 1 mM EDTA, 50 mM Tris–HCl, pH 7.6, 1 mM NaVO₄, 1 mM NaF, 1 mM DTT, 1 mM PMSF and Sigma protease inhibitor cocktail) for 20 min on ice. Whole tissue extracts were prepared by tissue homogenization in 1% NP-40 lysis buffer and incubation on ice for 15 min. Lysates were cleared by centrifugation at 13,000 rpm for 10 min. Protein was quantified with the Bio-Rad *DC* Protein Assay (Bio-Rad, Hercules, CA). 50 μg (10-well comb) or 1 mg (preparative comb) of protein was run out on 10% SDS-PAGE gels. Proteins were transferred to a nitrocellulose membrane and blocked with 5% bovine serum albumin or 5% non-fat milk in 1% Tween TBS (TBST). The following antibodies

were used for immunoblotting: mouse Bcl-2 (BD Pharmingen, San Jose, CA, Clone 3F11), human Bcl-2 (BD Pharmingen, Clone 6C8), or β-actin (Sigma, St. Louis, MO). For sera immunoblots, membranes were placed in a Surf-Blot apparatus (#5055, Idea Scientific, Minneapolis, MN) and probed with sera diluted at 1:200 in 5% non-fat milk in TBST for 2 hr at room temperature. Membranes were washed with TBST then incubated with an HRP-conjugated sheep anti-mouse IgG antibody (1:5000; GE Healthcare, Pittsburgh, PA) for 45 min and visualized with SuperSignal West Pico Chemiluminescent Substrate (Thermo Fisher Scientific, Waltham, MA) by autoradiography.

### In Vitro culture and TCR stimulation of thymocytes and T cells

Lymphoid organs were dissociated through 40 μm strainers into RPMI media (L-glutamine, Sodium Pyruvate, Non-essential amino acids, Hepes, β-mercaptoenthanol and Penicillin-Streptomycin). For assessment of TCR-mediated apoptosis, thymocytes were plated at $2 \times 10^6$ cells/well in triplicate in 96-well flat-bottom plates. Cells were left untreated or stimulated with either plate-bound anti-CD3 and anti-CD28 antibodies (clones 2C11 and PV-1, UCSF Hybridoma Facility) at the indicated concentrations for 18 hr. Death was assayed by Annexin V staining as described above. For assessment of cytokine production by splenic T cells, total splenocytes were stimulated for 4 hr with PMA and Ionomycin at the above concentrations in the presence of BD GolgiPlug per the manufacturer's protocol (Brefeldin A, BD Biosciences). To assess GFP downregulation in Nur77$^{GFP}$ cells, single cell thymocyte suspensions and mature T cells, purified by negative selection columns (T Cell Enrichment Columns, R&D Systems, Minneapolis, MN), were plated in 48-well plates at $2.5 \times 10^6$ cells/well and $1 \times 10^6$ cells/well, respectively. GFP expression by live cells was assessed every 24 hr over a 96 hr time course by flow cytometry.

### F5 and HY negative selection assays

For F5 in vivo peptide injection studies, F5 TCR transgenic mice were injected intraperitoneally with 50 nmol of Influenza NP$_{366-374}$ Strain A/NT/60/68 peptide (AnaSpec, Fremont, CA) in 200 μl of PBS or PBS only for controls. A second injection was performed 24 hr after the first. Thymocytes were collected at 48 hr after the initial injection.

For F5 and HY thymic slice assays, tissue preparation was performed as previously described (*Dzhagalov et al., 2012*, *2013*). In brief, thymic lobes were embedded in 4% GTG-NuSieve Low-melt Agarose (Lonza, Walkersville, MD) in HBSS. 500 μm slices were cut by Vibratome (1000 Plus, Leica, Buffalo Grove, IL) and placed in 0.4 μm Cell Culture Inserts (BD Biosciences) in 6-well plates containing 1 ml of RPMI media. TCR-specific (NP$_{366-374}$ Strain A/NT/60/68 and Smcy HY$_{738-746}$, AnaSpec) and control (NP$_{366-374}$ Strain A/PR/8/35 and OVA$_{257-264}$, AnaSpec) peptides were diluted to 100 ng/ml in RPMI media and 1 ml volume was added to the slices. Slices were incubated at 37°C in a plastic bag filled with 80% $O_2$ + 15% $N_2$ + 5% $CO_2$ (Blood Gas, Praxair, Danbury, CT) for 30 min. The peptide was then removed and slices were further incubated for the indicated times. Thymic slices were dissociated to create single cell suspensions and cell death was assessed by Cleaved Caspase 3 staining as described above.

### Mixed lymphocyte reaction

Balb/c (allogeneic) and C57BL/6 (syngeneic) splenocytes were harvested as described above and subjected to 2000 rads of γ-irradiation. Splenocytes were washed with RPMI media several times and plated at $1 \times 10^5$ cells/well in a round-bottom 96-well plate. T cells (C57BL/6 responders) were purified from lymph nodes and spleen by negative selection columns (T Cell Enrichment Columns, R&D Systems) or for T$_{reg}$-depleted MLRs by staining with PE-conjugated antibodies to CD25, CD19, CD11b, CD11c and Ter-119 (eBioscience and BD Biosciences) and negative selection using an EasySep Immunomagnetic PE Positive Selection Kit (StemCell Technologies, Vancouver, BC, Canada). The purified T cells were then labeled with 1 μM Cell Trace CFSE (Invitrogen, Life Technologies, Grand Island, NY) according to the manufacturer's protocol and seeded on top of stimulators at $1 \times 10^5$ cells/well in triplicate. Proliferation was assessed by CFSE dilution after 4 days.

### Anti-nuclear antibody ELISA

Serum anti-nuclear antibodies were quantified using a Mouse Anti-Nuclear Antibody Total Ig kit (Alpha Diagnostic International, San Antonio, TX) per the manufacturer's protocol. Briefly, samples were diluted at 1:100 in Low NSB Sample Diluent and added to a 96-well plate pre-coated with purified ENA. Quantification was performed relative to provided standards. The plate containing diluted sera and standards was incubated for 1 hr at room temperature. Anti-mouse Ig HRP was added for 30 min

followed by TMB substrate for 15 min. The reaction was stopped and the plate was read at 450 nm. Absorbance at 630 nm was subtracted as background.

## H&E staining

Organs were fixed in formalin, embedded in paraffin, and sliced into 5 μm sections by microtome. Sections were stained with Hematoxylin and Eosin (Thermo Fisher Scientific) by standard procedures. Images were captured with LAS Core V4.0 software (Leica) on a Leica DFC500 microscope equipped with a Leica DM2500 camera at 20x magnification (HCX PL-FLUOTAR 20X/0.50 objective, Leica).

## Acknowledgements

The authors would like to thank Ellen Robey, Heather Melichar, Ivan Dzhagalov and Jenny Ross for their insightful discussions on negative selection and instruction in the thymic slice model and additionally Ivan Dzhagalov and Ellen Robey for their critical review of the manuscript. We also thank Chulho Kang for technical assistance in creation of the BH3 Tg mice and April Choi and Erin Dunn for the extensive help with mouse husbandry.

## Additional information

### Funding

| Funder | Grant reference number | Author |
| --- | --- | --- |
| National Institutes of Health | CA66236 | Astar Winoto |
| Cancer Research Coordinating Committee | | Megan L Burger |
| National Institutes of Health | CA168007 | Megan L Burger |

The funders had no role in study design, data collection and interpretation, or the decision to submit the work for publication.

### Author contributions

MLB, Conception and design, Acquisition of data, Analysis and interpretation of data, Drafting or revising the article; KKL, MJB, Acquisition of data, Drafting or revising the article; AW, Conception and design, Analysis and interpretation of data, Drafting or revising the article

### Ethics

Animal experimentation: This study was performed in strict accordance with the recommendations in the Guide for the Care and Use of Laboratory Animals of the National Institutes of Health. All animal related procedures were carried out according to protocols (#RO33) approved by the institutional animal care and use committee of the University of California at Berkeley.

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
