## [Decision Letter]

Thank you for sending your work entitled “T cell-specific expression of a Bcl-2 BH3 domain mutant transgene inhibits negative selection and causes autoimmunity” for consideration at *eLife.* Your article has been favorably evaluated by Tadatsugu Taniguchi (Senior editor), a Reviewing editor, and 2 reviewers.

The Reviewing editor and the two reviewers discussed their comments before reaching a decision, and the Reviewing editor has assembled the following comments to help you prepare a revised submission.

In this manuscript, the authors use a novel reagent, a mutated form of Bcl2, to interfere with thymocyte clonal deletion. The idea is that this form will be more effective at blocking negative selection and will more definitively address the need for central tolerance. Increased expression of a Nur77-promoter/enhancer-driven GFP transgene indicated that thymocytes that received stronger TCR signals accumulated in the thymus. A thorough characterization of the Bcl-2 BH3 mice as compared with mice over-expressing Bcl-2 alone revealed no substantial differences. Yet, as the mice aged (>30 weeks old), they became ill and, sometimes, died. These mice had large spleens and more activated T cells.

On the whole the authors' data, especially with regard to CD8+ T cells, do back up their conclusion. The experiments are well done and the data are clearly presented. Expression of the transgene clearly protects thymocytes from various types of activation-induced cell death. However, the following points need to be dealt with in a revised version.

Major concerns:

1) An important concern with these studies is that the primary interpretation of the data, that disrupted negative selection leads to autoimmunity, is correlative. The most significant outcome, which is the onset of illness in old mice, could, arguably, be the result of many factors other than failed negative selection. Definitive proof of a one-to-one correlation in a system such as this (in other words, a T cell that escaped negative selection causes autoimmunity) is very difficult to prove. Indeed, in this model, it is not even clear whether the described disease is the direct result of T cell mediated autoimmunity. The authors need to discuss such caveats, in particular potential impacts of peripheral tolerance.

2) It is disappointing that no target of the autoimmune attack can be identified. The authors should use serum Igs to probe tissue sections or extracts for autoantibodies against targets other than nuclear antigens, the only targets tested in the manuscript. The authors might also get a bigger hit from their CD4+ MLRs if they were to use CD4+ T cells that are depleted of Tregs.

3) The primary argument used to support a failure of negative selection is that GFP expression from a Nur77 transgene is elevated. Nur77/GFP expression from this transgene is reported to directly report TCR mediated signaling. Consistent with this notion, there are a large number of thymocytes that express high levels of GFP in the presence of the BCL-2 transgenes. Oddly, the GFP expression continues to be elevated in mature thymocytes and even in peripheral T cells. Is the interpretation that the ligands that induce negative selection in the cortex are also present in the medulla and the spleen/lymph nodes? Or is it possible that Nur77 is upregulated for reasons not associated with the TCR? One simple experiment would be to make single cell suspensions of the thymocytes (remove them from MHC), incubate and show that GFP expression is reduced. Alternatively, bone marrow chimeric mice can be generated with Bcl-2/Nur77-GFP bone marrow and MHC deficient host mice.

---

## [Author Response]

*1) An important concern with these studies is that the primary interpretation of the data, that disrupted negative selection leads to autoimmunity, is correlative. The most significant outcome, which is the onset of illness in old mice, could, arguably, be the result of many factors other than failed negative selection. Definitive proof of a one-to-one correlation in a system such as this (in other words, a T cell that escaped negative selection causes autoimmunity) is very difficult to prove. Indeed, in this model, it is not even clear whether the described disease is the direct result of T cell mediated autoimmunity. The authors need to discuss such caveats, in particular potential impacts of peripheral tolerance*.

We agree that it is difficult to prove that T cells that have escaped negative selection are directly causing autoimmunity; however, we believe we offer strong correlative evidence. The generally low autoantibody titers against ANA along with the early activated T cell phenotype and multi-organ autoantibody response (see response to point 2 below) argue in favor of T cell-mediated autoimmunity rather than a systemic, SLE-like disease. A defect in peripheral T cell deletion due to increased T cell survival may also contribute to the BH3 Tg phenotype. Although we cannot completely rule out this possibility, the fact that Bcl-2 Tg mice do not develop autoimmunity would suggest a minimal role of peripheral tolerance in this model. We have included a statement concerning this point in the revised discussion. In addition, we have softened our language throughout the text to reflect the caveat that factors other than negative selection may play a role in the autoimmunity of BH3 Tg mice.

*2) It is disappointing that no target of the autoimmune attack can be identified. The authors should use serum Igs to probe tissue sections or extracts for autoantibodies against targets other than nuclear antigens, the only targets tested in the manuscript*.

We have performed the suggested experiment and presented the findings in the new Figure 8. This figure depicts tissue extracts from non-immune organs (pancreas, eye, lung, liver, kidney and stomach) probed with sera from aged mice. Interestingly, autoantibody binding was detected for all the BH3 Tg mice examined and most if not all the mice had autoantibodies against multiple tissues. Common banding patterns were observed amongst the BH3 Tg mice within a given tissue suggesting common autoantibody targets. Occasionally, we also detected autoantibody binding in some Bcl- 2 Tg mice, but fewer or weaker bands were observed compared to the BH3 Tg mice. We also detected an occasional band in one of the wild-type controls, maybe due to the old age of the mice, but these bands did not correspond with the banding patterns observed in the BH3 Tg mice. The presence of autoantibodies in all the BH3 Tg mice that react to multiple bands in the tissue extracts further support our conclusion that the BH3 Tg mice develop multi-organ autoimmunity.

*The authors might also get a bigger hit from their CD4+ MLRs if they were to use CD4+ T cells that are depleted of Tregs*.

The new Figure 7 is a representative T_reg_-depleted MLR. However, we did not observe an improvement of CD4 T cell proliferation with T_reg_ depletion.

*3) The primary argument used to support a failure of negative selection is that GFP expression from a Nur77 transgene is elevated. Nur77/GFP expression from this transgene is reported to directly report TCR mediated signaling. Consistent with this notion, there are a large number of thymocytes that express high levels of GFP in the presence of the BCL-2 transgenes. Oddly, the GFP expression continues to be elevated in mature thymocytes and even in peripheral T cells. Is the interpretation that the ligands that induce negative selection in the cortex are also present in the medulla and the spleen/lymph nodes? Or is it possible that Nur77 is upregulated for reasons not associated with the TCR? One simple experiment would be to make single cell suspensions of the thymocytes (remove them from MHC), incubate and show that GFP expression is reduced. Alternatively, bone marrow chimeric mice can be generated with Bcl-2/Nur77-GFP bone marrow and MHC deficient host mice*.

Moran et. al [JEM 2011] have previously showed that Nur77 GFP expression correlates absolutely with TCR signaling strength. GFP signal is gradually downregulated as thymocytes mature from DP to SP to recent thymic emigrant and naïve T cells in the periphery. They showed that naïve T cells do continue to express a low level of Nur77 GFP because naive T cells presumably need a low level tonic TCR stimulation to survive (as demonstrated by several classical papers) . Indeed, CD4 T cells transferred to an MHC II-deficient host downregulated GFP further over time (Moran et al. JEM 2011). Hence, Nur77 GFP expression in mature T cells is likely in part leftover from TCR engagement in the thymus, but is also due to TCR engagement in the periphery.

The Nur77 GFP expression observed in BH3 and Bcl- 2 Tg T cells also appears to be TCR-dependent, and not transgene-dependent, as transgenic T cells downregulated GFP at a similar rate to WT T cells when removed from MHC in culture (new Figure 5—figure supplement 1). Since SP thymocytes continue to undergo negative selection in the medulla, increased Nur77 GFP expression by BH3 and Bcl-2 Tg SP cells can be due to both rescue of cells seeing ligand in the medulla and leftover expression from stimulus at the DP stage. Similar to mature T cells, we found that SP thymocytes removed from MHC by culturing, downregulate GFP at a similar rate to WT SP thymocytes, suggesting that GFP expression is TCR-dependent, and not transgene-dependent (new Figure 3—figure supplement 1). In addition, we have shown in the original manuscript that transgenic pre-selection DP thymocytes do not express Nur77 GFP (Figure 3), suggesting again that Nur77 GFP expression is TCR -dependent. As bone marrow chimera-derived thymocytes in an MHC- deficient host would be arrested at the pre-selection DP stage, we don’t believe this experiment would provide any additional information.